# Successful introgression of *w*Mel *Wolbachia* into *Aedes aegypti* populations in Fiji, Vanuatu and Kiribati

Cameron P. Simmons[1]*, Wesley Donald[2], Lekon Tagavi[2], Len Tarivonda[2], Timothy Quai[3], Raynelyn Tavoa[4], Tebikau Noran[5], Erirau Manikaoti[5], Lavinia Kareaua[5], Tabomoa Tinte Abwai[5], Dip Chand[6], Vineshwaran Rama[6], Vimal Deo[6], Kharishma Karti Deo[6], Aminiasi Tavuii[1], Wame Valentine[7], Raviikash Prasad[7], Eremasi Seru[7], Leikitah Naituku[7], Anaseini Ratu[1], Mark Hesketh[1], Nichola Kenny[1], Sarah C. Beebe[1], Anjali A. Goundar[1], Andrew McCaw[1], Molly Buntine[1], Ben Green[1], Tibor Frossard[1], Jeremie R. L. Gilles[1], D. Albert Joubert[1], Geoff Wilson[1], Le Quyen Duong[1], Jean B Bouvier[1], Darren Stanford[1], Carolyn Forder[1], Johanna M. Duyvestyn[1], Etiene C. Pacidônio[1], Heather A. Flores[1], Natalie Wittmeier[1], Kate Retzki[1], Peter A. Ryan[1], Jai A. Denton[1], Ruth Smithyman[1], Stephanie K. Tanamas[1], Peter Kyrylos[1], Yi Dong[1], Anam Khalid[1], Lauren Hodgson[1], Katherine L. Anders[1], Scott L. O'Neill[1]

**1** World Mosquito Program, Monash University, Clayton, Australia, **2** Ministry of Health, Government of Vanuatu, Port Vila, Vanuatu, **3** World Mosquito Program, Port Vila, Vanuatu, **4** Vanuatu Red Cross Society, Port Vila, Vanuatu, **5** Ministry of Health and Medical Services, Kiribati Government, Kiribati, **6** Ministry of Health and Medical Services, Government of Fiji, Suva, Fiji, **7** World Mosquito Program, Suva, Fiji

\* cameron.simmons@worldmosquito.org

**Data Availability Statement:** Raw data is available on figshare at doi.org/10.6084/m9.figshare. 24720948.

## Abstract

Pacific Island countries have experienced periodic dengue, chikungunya and Zika outbreaks for decades. The prevention and control of these mosquito-borne diseases rely heavily on control of *Aedes aegypti* mosquitoes, which in most settings are the primary vector. Introgression of the intracellular bacterium *Wolbachia pipientis* (*w*Mel strain) into *Ae. aegypti* populations reduces their vector competence and consequently lowers dengue incidence in the human population. Here we describe successful area-wide deployments of *w*Mel-infected *Ae. aegypti* in Suva, Lautoka, Nadi (Fiji), Port Vila (Vanuatu) and South Tarawa (Kiribati). With community support, weekly releases of *w*Mel-infected *Ae. aegypti* mosquitoes for between 2 to 5 months resulted in *w*Mel introgression in nearly all locations. Long term monitoring confirmed a high, self-sustaining prevalence of *w*Mel infecting mosquitoes in almost all deployment areas. Measurement of public health outcomes were disrupted by the Covid19 pandemic but are expected to emerge in the coming years.

## Author summary

For decades, dengue, Zika and chikungunya have been public health issues across the Pacific Island region. *Aedes aegypti* mosquitoes are considered most responsible for the transmission of dengue between people. The introduction of a bacteria called *Wolbachia pipientis* (*w*Mel strain) to these mosquitoes is known to reduce the transmission of these diseases. Herein, we describe the production and release of *w*Mel-carrying *Ae. aegypti* mosquitoes

**Funding:** This work was supported by a grant award from the Commonwealth of Australia represented by the Department of Foreign Affairs and Trade titled; Operational pilot of Wolbachia technology to reduce the transmission of Aedes aegypti-borne diseases ("the Project") awarded to SLO. The following authors received full or part salary from this grant for the work described here; WD, LT, TQ, RT, TN, EM, LK, TTA, VR, AT, WV, RP, ES, LN, AR, MH, NK, SCB, AGB, AM, MB, GW, LQD, JBB, DS, CF, NW, KR, PAR, JAD, RS, SKT, PK, YD, AK, LH. The funders had no role in study design, data collection and analysis, decision to publish, or preparation of the manuscript.

**Competing interests:** The authors have declared that no competing interests exist.

into several Pacific Island cities, including Suva, Lautoka, and Nadi in Fiji, Port Vila in Vanuatu, and South Tarawa in Kiribati. With community support, these mosquitoes were released on a weekly basis for periods ranging from 2 to 5 months. The result was a widespread integration of the *w*Mel bacteria into local mosquito populations. Long-term monitoring has shown that the *w*Mel bacteria has been sustained at high levels in mosquitoes in nearly all of the areas where it was introduced. This innovative approach could potentially improve the way we combat mosquito-borne diseases, protecting communities in the Pacific Islands and beyond from the devastating effects of dengue, chikungunya, and Zika.

## Introduction

Arboviral diseases, like dengue, chikungunya, and Zika are a public health threat in over 100 countries. Dengue was first reported in Pacific Island countries (PICs) in the mid-19th century [1], and periodic major outbreaks of dengue have occurred since the 1970s [2]. Chikungunya emerged in multiple PICs in 2011 [3], soon followed by Zika in 2013 [4]. Between January 2012 and September 2014, an unprecedented 28 arboviral outbreaks were reported from the Pacific: 18 dengue, 7 chikungunya and 3 Zika [4]. Arboviral outbreaks have continued since [5]. The yellow fever mosquito, *Aedes aegypti*, is a key vector of these arboviral diseases and thus a target of public health control strategies.

Dengue control and prevention efforts have relied mostly on suppressing the population of *Ae. aegypti*, but this has proven hard to sustain and has not demonstrably eliminated the threat of dengue outbreaks in any dengue endemic country. A dengue vaccine was authorised for use in Indonesia and the European Union in 2022 [6], but is yet to be deployed programmatically in any endemic country. The *Wolbachia pipientis* (*Wolbachia*) introgression method has emerged as a novel public health intervention for control of arboviral diseases. This obligate intracellular bacterium naturally occurs in many insect species [7–9] but is not naturally present in *Ae. aegypti* [10,11]. When stably transinfected with the *w*Mel strain of *Wolbachia*, *Ae. aegypti* have reduced vector competence for dengue virus (DENV) types 1–4, Zika and chikungunya viruses [12–16]. This reduced vector competence phenotype can be durably established into an *Ae. aegypti* population because *w*Mel manipulates reproductive outcomes to favour its own population introgression [17].

Stable introgression of *w*Mel into *Ae. aegypti* populations has been demonstrated in Australia, Asia and Latin America [18–23] and an accumulating body of evidence from randomised and non-randomised trials has reproducibly demonstrated reductions in dengue incidence after *w*Mel introgression [19–23]. Reductions in chikungunya and Zika incidence have also been reported in Brazil [20]. In December 2020, the World Health Organisation's (WHO) Vector Control Advisory Committee (VCAG) concluded there was sufficient evidence for WHO to initiate the guideline development process [24], a process that has commenced.

We describe here successful deployments of *w*Mel-infected *Ae. aegypti* in the Pacific Island countries of Fiji (Suva, Lautoka, Nadi), Vanuatu (Port Vila) and Kiribati (South Tarawa), and the approaches available for evaluating the long-term public health benefits in each location.

## Methods

### Ethics statement

Monash University Human Research Ethics (permit #3093) approved the blood feeding of mosquitoes on the on the arms or legs of consenting adult human volunteers (residents of

**Table 1. Target area size and population for *Wolbachia*(*w*Mel)-infected *Ae. aegypti* deployments in Fiji, Vanuatu and Kiribati.**

| Country | Project Sites | Target Area (km$^2$) | Target Population (2021) |
|---|---|---|---|
| Fiji | Suva, Lami, Nausori, Nadi, Lautoka | 116.4 | 343,704 |
| Vanuatu | Port Vila | 38.7 | 61,297 |
| Kiribati | South Tarawa | 5.3 | 59,275 |

Melbourne, Australia where *Ae. aegypti* is absent and where arboviruses like dengue, Zika and chikungunya do not circulate). All volunteers provided written informed consent to bloodfeed mosquitoes.

### Intervention areas

Planned areas for deployment of *w*Mel-infected *Ae. aegypti* releases in Fiji, Vanuatu and Kiribati covered a total of 464,276 people across 160.4 km$^2$ (Table 1). Deployments were focused in urban areas with high population densities. In Fiji, these areas were the Lami-Suva-Nasinu corridor in the Central Division, and Lautoka and Nadi in the Western Division. In consultation with the Fiji Ministry of Health and Medical Services (MHMS), each urban area was divided into operational reporting areas for the purposes of *w*Mel deployments. Port Vila is the largest city and the capital of Vanuatu and the twelve administrative areas chosen for release of *w*Mel-infected mosquitoes were determined together with the Vanuatu Ministry of Health, and comprised urban (residential and commercial) areas in and around Port Vila. In Kiribati, more than half the population resides in half the population resides in South Tarawa. *w*Mel-infected mosquito deployments were planned for five areas in South Tarawa—Ambo, Bairiki, Betio, Bikenibeu Causeway and Teaoraereke-Ambo.

### Partnerships

In Fiji, WMP partnered with the non-governmental organisation (NGO) Live and Learn Environmental Education and the Fijian government through the Ministry for Health and Medical Services. In Vanuatu, WMP partnered with the Vanuatu Red Cross and the Vanuatu Government through the Ministry of Health. In Kiribati, WMP partnered with the Government of Kiribati through the Ministry for Health and Medical Services.

### Regulation

Regulatory approvals were obtained from corresponding local authorities in each country. In each case, deployment required exporting locally caught *Ae. aegypti* from target countries and importing these mosquitoes into Australia for use at the WMP Monash University production facility. Once generated, localised *w*Mel-infected *Ae. aegypti* lines were exported from Australia and imported into the corresponding release country. Therefore import, export and facility permits were required. In the case of Kiribati, a Fijian transit permit was also required due to lack of a direct flight. Details of relevant permits and their corresponding country are provided (S1 Table).

### Community engagement

In each country a communications and engagement team implemented the WMP's Public Acceptance Model (PAM) for obtaining community support for mosquito releases as described previously [19,25]. Briefly, the goal of communication and engagement was: a)

awareness, where the majority of a release community was aware of the *Wolbachia* method; b) support, where the majority accepted WMP releases; and, c) participation, where residents were willing to house mosquito traps and release boxes or were advocating for the *Wolbachia* method with others. This consisted of four components:

1. Raising awareness. Information about the program and our activities was provided to the community and other stakeholders via a broad range of channels, including radio, newspapers, social media, street banners, market stalls, community door-knocking, school outreach programs and announcements via churches and chiefs.

2. Quantitative surveys to assess community awareness and support were conducted by the team at the start of community engagement (baseline survey) and before mosquito releases (pre-release survey). Surveys were conducted face-to-face, at every fifth house from randomly selected starting points across the areas, with one respondent per house.

3. An issues management system was established to allow staff to track community questions or concerns, and to reply to them within 24 hours. Community members could also choose to opt out of direct participation in the release or monitoring activities.

4. A community reference group (CRG) was established, representing the diversity of stakeholders within the communities. Members included representatives from local government, businesses, the tourism industry, police force, chiefs and the community. The function of the CRG was to independently review deployment activities to ensure that engagement was carried out in accordance with our stated Public Participation Principles.

**Fiji.**    Communication and engagement occurred in two stages: phase one focused on the Suva, Lami and Nausori area from April to July 2018, and phase two focused on Nadi and Lautoka from January to March 2019. Engagement in phase one included the distribution of over 26,000 *Wolbachia* method fact sheets to local mailboxes, publication of five newspaper articles, six radio/television shows and sponsorship of the Coca-Cola Games. Viewership of the Games was estimated to be 576,000 people. Paid advertising in English, iTaukei and Hindi occurred on two television networks and four radio stations. WMP also established a total of nine face-to-face community events, one launch and eight awareness sessions, to allow members of the community to engage with the program. Social media posts explaining the method were viewed by 74,639 people. Prior to pre-release acceptance surveys, in June and July 2018, additional community engagement was undertaken: specifically, three radio programs, two in iTaukei and one in Hindi, two television segments, one in English and one Hindi, direct mail to 26,227 letter boxes, plus face-to-face engagement in Vatuwaqa, Nakasi, Lami and Raiwaqa which were areas of particularly low awareness.

Beginning in January 2019, phase two of the community engagement in Nadi and Lautoka followed a similar strategy to phase one. This included 16 published media articles, 16,500 direct mail outs, 19 face-to-face engagement events and 13 social media posts. Baseline acceptance surveys were not undertaken as potential participants would potentially be aware of the release from phase one engagement. However, pre-release surveys were undertaken in both Nadi and Lautoka.

Two independent CRGs were established for the Fijian deployments. The first covering the Suva, Lami and Nausori area consisting of 12 members of the local community and the second covering Nadi and Lautoka consisting of five community members.

**Vanuatu.**    Like Fiji, communication and engagement in Vanuatu employed diverse media streams to ensure the largest number of residents were reached. From September 2017 until

**Table 2. Public Acceptance Surveys.** Outcome of community surveys taken at baseline (prior to engagement activity) and pre-release (prior to deployment of *Wolbachia*-infected mosquitoes). Awareness was determined by asking participants if they had heard of the World Mosquito Program. Acceptance was determined by asking participants if, once it was explained to them, they approved of releasing mosquitoes with good bacteria to reduce dengue.

| Country | Baseline number of participants | Baseline Acceptance | Baseline Awareness | Pre-release number of participants | Pre-release Acceptance | Pre-release Awareness |
|---|---|---|---|---|---|---|
| Fiji–Suva | 402 | 98% | 27% | 405 | 98% | 42% |
| Fiji–Nadi | – | – | – | 319 | 97% | 87% |
| Fiji–Lautoka | – | – | – | 217 | 96% | 63% |
| Kiribati | 298 | 97% | 33% | 299 | 95% | 64% |
| Vanuatu | 301 | 96% | 27% | 305 | 93% | 56% |

November 2018, there were a total of 38 radio or newspaper pieces on the topic of deploying *Wolbachia*-infected mosquitoes. All of these had a positive tone and were supportive of the deployment. In addition, there were 104 info stalls/sessions, 23 presentations to particular community groups, and 21 events throughout the project marking key milestones including Vanuatu WMP project launch and the release of the first adult mosquitoes. Finally, WMP paid for advertising that included over 60,000 messages to local mobile phones, three paid radio adverts and two paid TV advertisements.

The Vanuatuan CRG consisted of 20 community representatives. This included individuals from the local and federal government, the police force, local business, the tourism industry and special interest groups.

**Kiribati.** Communication and engagement in Kiribati relied on a CRG and local media. This included four newspaper articles, eight radio events and one radio advertisement. In addition 400 posters and four banners were displayed in stores, medical clinics and motels. A total of 1,263 pamphlets were distributed. There were also two stalls established that 250 people visited and 61 community awareness events that were attended by 3,261 people.

In Kiribati the CRG consisted of 13 members. Once again, careful selection with local partners was undertaken to ensure diverse representation. This included members of numerous community groups, religious groups and non-government organisations.

**Outcomes.** In each country, WMP communications and engagement increased the percentage of participants aware of the *Wolbachia* method by 15–31 percentage points compared to the baseline survey result (Table 2). However, engagement provided no additional increase to community acceptance levels as baseline acceptance was already greater than 90 percent in each country (Table 2).

After presentation of all collected data to each site-specific CRG, their endorsement was required prior to commencing the deployment of *w*Mel-infected *Ae. aegypti*. The Fijiian CRG provided endorsement for mosquito releases to proceed in June 2018 for the Suva, Lami and Nausori deployment (phase1) and April 2019 for the Nadi and Lautoka deployment (phase 2). The CRGs in Vanuatu and Kiribati provided endorsement for mosquito releases to commence in May 2018.

## Mosquito production

WMP generated *w*Mel-infected Fiji, Vanuatu and Kiribati *Ae. aegypti* lines, Fij-*w*Mel, Van-*w*Mel, and Kir-*w*Mel, via backcrossing, performed quality assurance, and mass-produced mosquito eggs in Melbourne, Australia. Eggs were shipped to either Fiji, Vanuatu or Kiribati where mosquitoes were reared to adults for release by local staff.

**Backcrossing.** The Fij-*w*Mel, Van-*w*Mel, and Kir-*w*Mel lines were created by rearing and backcrossing field-derived *Ae. aegypti* males from Suva, Port Vila and South Tarawa respectively with females from the Cairns *Ae. aegypti* *w*Mel-infected line for six generations, as previously described [26,27]. Briefly, ovitraps were used to acquire *Ae. aegypti* eggs from field locations and the latter were then hatched in the laboratory where *Ae. aegypti* were visually identified, caged and then blood fed. The F1 eggs from these field-caught *Ae. aegypti* were collected and shipped to Monash University, Melbourne, Australia where the backcrossing took place. The blood and field-derived F1 adults were confirmed as being test-negative for medically important flaviviruses and alphaviruses at the Victorian Infectious Diseases Reference Laboratory (VIDRL) before eggs were hatched.

**Fitness assays.** At the end of backcrossing a standard panel of quality assurance tests were performed on the Fij-*w*Mel, Van-*w*Mel, and Kir-*w*Mel lines to ensure they met quality criteria. These included–a) assessments of the fidelity of maternal transmission of *w*Mel (must be greater than 90%), b) the degree of cytoplasmic incompatibility induced (must be greater than 90%) (S2 Table), c) average fecundity per female (S2 Table), vector competence (S1 Fig; S3 Table) (a significant reduction in DENV RNA copies per mosquito), and an insecticide resistance phenotype consistent with the country's wild-type mosquitoes. The insecticide resistance phenotype of each *w*Mel line, and their corresponding wild-type mosquitoes, was measured according to the World Health Organisation bioassay method [28] (S2 Fig). Maternal transmission was determined by crossing *w*Mel-infected females with field-derived uninfected males (e.g. *w*Mel-Kir females crossed with uninfected males derived from Kiribati egg collections) as previously described [29]. Cytoplasmic incompatibility was determined by crossing uninfected females, derived from the corresponding country, with *w*Mel-infected males as previously described [29]. In both instances, individual mated females were raised in iso-female tubes and eggs laid and larvae hatched were determined. When possible, 30 larvae were raised to four-to-six day old adults and tested for *w*Mel infection. The vector competence of each *w*Mel line for the four DENV serotypes was determined as described in Pocquet et al [29] using intrathoracic injection. The DENV RNA concentration in individual mosquitoes was determined by qPCR. We suggest that intrathoracic challenge is a reproducible approach to measuring the restriction on DENV replication in the mosquito body. We consider the parental *w*Mel Cairns line as the "gold standard" and expect that newly created lines will have similar phenotypes in response to intrathoracic injections with DENV-1-4.

**Mass production of eggs.** The "brood stock" of country-specific *w*Mel lines were maintained at Monash University, Australia, as previously described [19]. Briefly, 300–600 larvae were reared per 40 x 30 x 8 cm plastic tray with 2 L reverse-osmosis (RO) water, and fed as required on Tetramin Tropical Tablets (Tetra Holding [US] Inc. Germany, product number 16110) or Aqua One Vege Wafers (Aqua Pacific UK Ltd, Southampton, UK, product number 26050). Pupae were then transferred to and emerged in mesh cages (30 x 30 x 30 cm) (WMP), at a density of 600 mosquitoes per cage. Ten percent wild-type males were introduced each generation.

The prevalence of *w*Mel-infected females was tested using qPCR in each generation. For mass production, eggs were hatched into 40 x 30 x 8 cm trays (Kartell Ltd, Australia), with 3 L of RO water, to achieve larval densities of 3500–3800 per tray. Larvae were initially fed on Tetramin Tropical Tablets: for days 1–6, sequentially, 0.5, 1, 5, 9, 9, and 4 tablets were added per tray. When Tetramin was not available, a liquid diet (37.5 g tuna meal, 26.3 g beef liver powder and 11.3 g baker's yeast, made up to 1 L in RO water) was used. The volumes of liquid diet used on days 1–6, sequentially, were 3, 10, 15, 30, 30, and 10 mL per tray. A sample of third instars from each tray was tested for *w*Mel prevalence.

Pupae were transferred to large mesh cages (90 x 90 x 20 cm) (WMP), for an adult stocking density of 12,000–15,000 mosquitoes per cage. Mosquitoes were maintained on sucrose solution with 0.4% propionic acid, and fed on human blood (Red Cross, Australia) using an artificial feeding apparatus for three gonotrophic cycles. Eggs were laid on strips of seed germination paper (Sartorius AG, Germany, product number FT-2-314-580580).

**Egg transportation.** *w*Mel-infected *Ae. aegypti* eggs (typically less than 2 weeks old) were transported weekly from the WMP insectary facility in Melbourne, Australia to Fiji, Vanuatu and Kiribati, as described previously [30]. Briefly, each batch of egg strips was packed in a temperature and humidity controlled box and transported by the specialist courier company, Lab-Cabs International. Damp cotton wool balls placed in an open Ziploc bag were included in the box to maintain humidity. The average egg hatch rate after shipment from Australia was greater than 50 percent in each of the release countries.

**Production of adult mosquitoes for release.** On arrival at the rearing facilities, eggs were brushed from the egg strips. Hatch rate was estimated by placing 400 eggs in a small vial of hatching solution– 2 crushed Tetramin Tropical Tablets in 1 L of boiled water–and counting unhatched eggs 3 hours later.

Remaining eggs were then weighed, and sufficient eggs to produce approximately 25,000 larvae were hatched and transferred into each large rearing tray (117 x 35 x 8 cm). Immatures were fed on the liquid diet described above. For 25,000 larvae, the following volumes of liquid diet were added to each tray for days 1–6 sequentially: 25 mL, 50 mL, 70 mL, 145 mL, 290mL, 200mL and 200mL. Trays were flushed and refilled with fresh water as needed.

From each egg shipment, a total of 160 mosquito larvae were collected from the mass rearing trays and placed in 2mL tubes containing 70–80% ethanol (w/w). Samples were shipped to Monash University, Melbourne, Australia, where they were tested using qPCR, as described below, to measure prevalence of *w*Mel infection.

Approximately five days after hatching, 150 pupae were transferred into individual release tubes. In Suva the release tubes were locally made by cutting PVC pipes to size (~15 x 5 cm). These Suva adult tubes were also transported to Nadi for releases there. This resulted in adult mosquitoes being kept in the tubes for 3–5 days. In Port Vila and South Tarawa, release tubes were 14 x 5 x 5 cm injection-moulded plastic mesh release tubes (iCreate Retail Solutions, Scoresby, Australia). The tubes were then placed in shallow trays of water. Adults were allowed to emerge and were maintained in the tube for 3–5 days before release. Cotton pads soaked in 25% sugar solution or honey were put on the tubes as needed for maintenance.

**Mosquito release containers.** Mosquito release containers (MRCs) are small 750 mL containers that facilitate the aquatic *Ae. aegypti* life cycle, from egg to adult [19,23]. Added to each container was approximately 400 ml of water, 100–150 *w*Mel-infected *Ae. aegypti* eggs and small amounts of mosquito larval food. Over 2 weeks, the eggs hatch, develop into adult mosquitoes and emerge from the MRC.

**Mosquito releases.** Deployments of *w*Mel-infected *Ae. aegypti* in Fiji, Vanuatu and Kiribati consisted of adult mosquitoes being released from a vehicle or on foot. In Kiribati, egg releases also occurred. In general, releases continued in each reporting zone until two or more consecutive measurements of the *w*Mel prevalence exceeded 50%. In a small number of zones this threshold was not reached before the end of the planned approved release period. Mosquito release dosages by area is available in S3–S6 Figs.

**Fiji.** Fijian releases occurred in two phases. Phase 1 releases were undertaken in Suva, Nausori and Lami between July 2018 and May 2019, and phase 2 releases in Nadi and Lautoka between May and November 2019. Suva, Nadi and Lautoka were divided into 12, six and five release zones, respectively, for operational purposes. Mosquito release locations were guided by overlaying a 100 m$^2$ grid over each release area, with one release point per grid square. In

most release points a single tube with approximately 100–150 mosquitoes was released, except for areas of high human population density where two tubes (~300 mosquitoes) were released. Releases were intended to be done weekly, but in practice not all points received releases of this frequency as production, shipping and/or rearing constraints limited the mosquitoes available. Therefore, we planned releases in a way where the sites that were left out previously were replenished first.

**Vanuatu.** Releases in Port Vila occurred between October 2018 and March 2019. The release area of 39 km$^2$ was divided into 12 release zones, congruent with public health reporting boundaries. Release points were evenly spaced across each release zone, on a grid of approximately 75 m$^2$. Adult mosquitoes were transported to the field by vehicle, then to release points either by vehicle or on foot, due to mixed terrain across the release area. At most release points, a single tube of approximately 150 adult mosquitoes were released. For a small number of areas of high human and/or mosquito density, two tubes of mosquitoes were released at each point.

Mosquito releases were progressively rolled out across different release zones, beginning with zones with higher human population density and reported dengue incidence in recent outbreaks. Again, releases were intended to be done weekly but in practice constraints on mosquito production, shipping and rearing meant that not all points within each release area received releases each week. The number of weeks in which a release occurred in each area, an estimation of total mosquitoes released, mosquitoes released per km$^2$ and mosquitoes released per inhabitant are provided in S4 Table.

**Kiribati.** Releases in South Tarawa occurred from August 2018 to August 2019. Initially South Tarawa was divided into eight release zones. Releases consisted of both adult mosquito releases from tubes and deployment of eggs in MRCs. The number of weeks in which a release occurred in each reporting area, an estimation of total mosquitoes released, mosquitoes released per km$^2$ and mosquitoes released per inhabitant are provided in S4 Table. During the early months of releases it became apparent that reliably producing enough *w*Mel-infected mosquitoes in Kiribati to meet release requirements was difficult. This was mainly because of insufficient eggs being delivered from the Melbourne insectary. It was therefore decided to reduce the geographic scope of releases (from eight to two reporting areas- Bairiki and Betio) in order to increase the probability of *w*Mel being established. Thus, release tubes containing adult *w*Mel-infected *Ae. aegypti* or MRCs containing egg capsules were redirected from other release zones to either Bairiki or Betio.

**Field monitoring.** In Suva (Fiji), Port Vila (Vanuatu) and South Tarawa (Kiribati) adult mosquitoes were collected from the field weekly during the release period using Biogents Sentinel 2 (BGS) traps (Biogents AG, Regensburg, Germany, Product number NR10030). In Suva and Port Vila traps were placed at a density of approximately 3–10 per km$^2$; whereas in South Tarawa this density was approximately 13–20 per km$^2$. Within 3–6 months of completing releases, additional periodic mosquito collections were done in each site to monitor *w*Mel introgression. Long term monitoring was interrupted in 2020 and 2021 by Covid-19 pandemic restrictions. Nonetheless, long-term monitoring was completed on three occasions in Suva (approximately 1, 2 and 3 years post-release) on two occasions in Nadi, Lautoka and Port Vila (approximately 1 and 2 years post-release) and once in Kiribati (approximately 2.5 years post-release). In the Phase 2 Fiji deployments in Nadi and Lautoka, and in periodic post-release monitoring in Suva, BG traps were supplemented with collections using handheld, battery-powered aspirators. The team was provided with a map that had predetermined clusters of houses circled where each cluster had five houses to be surveyed using Prokopacks. Typically there were between 2–5 clusters per km$^2$. The density of sampling in post-release monitoring was less than in the release phase, and for a shorter duration (2–4 weeks), and hence a lower yield of Ae. aegypti were caught.

Samples collected from BGS traps and aspirations were identified morphologically. All *Ae. aegypti* samples were stored in 70% ethanol and shipped to Monash University, Melbourne, Australia for diagnostic testing for the presence/absence of *w*Mel in each *Ae. aegypti*.

**Wolbachia diagnostic testing.** Adult mosquitoes collected from the field were screened for *w*Mel infection at Monash University, Melbourne, Australia. *w*Mel diagnostic testing was undertaken using either LAMP or PCR, as previously described [19,31]. The qPCR assay includes a primer/probe set to detect *Ae. aegypti*, while LAMP does not; qPCR testing thereby provided a quality check for the mosquito identification process. Mosquitoes from Vanuatu were tested using only qPCR, whereas mosquitoes from Fiji and Kiribati employed both LAMP and qPCR methods.

**Training, data storage & analysis.** WMP has developed customised web and mobile applications referred to as Core Data. Technologies used to develop the platform include Django, Python, Javascript and ODK-X applications. Core Data houses both entomological and epidemiological data from all global release sites. Data visualisation and site comparisons are facilitated through a series of data-dashboards. This framework ensures preservation of *Wolbachia* Method implementation data and facilitates ongoing operations.

Site-specific training in Fiji, Vanuatu and Kiribati was undertaken through a combination of deployment of experienced WMP staff, in person training and use of the WMP Catalyst system. Catalyst is an internally developed platform, built on the Fuse Universal (London, UK) acting as digital education and training platform.

Entomological data was exported from Core Data Analysis of field and *w*Mel introgression data was conducted using R through RStudio. Data was collated and visualised using several R packages including 'ggplot2', 'tidyverse', 'lubridate' and 'ggh4x'. Raw data is available on figshare at doi.org/10.6084/m9.figshare.24720948. Epidemiological data was analysed using Stata (StataCorp, Texas, USA) and visualised using GraphPad (Insight Partners, New York, USA).

**Description of public health impact.** The preliminary public health outcomes of the *Wolbachia deployments* were described using data on notified cases of dengue and other *Aedes*-borne diseases (where available) from the routine disease surveillance systems in each country.

**Vanuatu.** Suspected and laboratory-confirmed dengue cases are notified to the Surveillance Unit of the Vanuatu Ministry of Health. Comparable time series data is only available since 2016, before which time dengue surveillance was included within the Ministry of Health malaria program and there was no electronic data reporting. Dengue and chikungunya have similar clinical presentations, therefore during non-outbreak periods, clinically suspected cases of dengue and chikungunya are reported together as "dengue-and-chikungunya-like illness". Data on clinically suspected cases reported during non-outbreak periods is aggregated by week and reporting health facility, without information on neighbourhood of residence. For our purposes, all clinically suspected cases notified by Vila Central Hospital are assumed to be residents of Port Vila, though in reality these will include an unknown subset who reside outside of Port Vila, and thus outside of the *Wolbachia* release area, but who presented to Vila Central Hospital as it is the main referral hospital in Vanuatu.

Laboratory diagnostic testing for dengue is generally only performed during outbreak periods, and uses IgM and NS1 rapid tests. Until 2020, line-listed laboratory results including neighbourhood of residence were available for all suspected dengue cases regardless of their test result. Since 2020, only cases with a positive dengue IgM or NS1 rapid test result were included in the line-listed data; residential location was not available for dengue test-negative samples. No chikungunya or Zika laboratory data is available in Vanuatu, and no routine Zika surveillance is conducted in Vanuatu. For this analysis, dengue case notifications data was available for the period January 2016 to January 2022.

**Kiribati.** There is syndromic surveillance for dengue and chikungunya in Kiribati. An outbreak period is defined by the detection of one or more laboratory-confirmed dengue or chikungunya cases. Data for clinically suspected dengue and chikungunya cases is available from the KMHMS laboratory. This data consists of laboratory lists of suspected cases who had specimens sent for diagnostic testing and thus also includes information on results of laboratory testing for both dengue and chikungunya. The number of specimens listed for testing per month represents the total number of suspected cases. Although the database includes residential location, this field is not reliably reported and so data may also include cases from areas of South Tarawa where *Wolbachia* deployments were not performed. There is no Zika diagnostic testing in Kiribati. For this analysis, dengue case notifications data was available for the period January 2009 to August 2022, and chikungunya case notifications from January 2017 to August 2022.

**Fiji.** In Fiji, three surveillance systems within the Ministry of Health and Medical Services (MOHMS) report independently on suspected and confirmed dengue cases: National Notifiable Disease Surveillance System (NNDSS), the Early Warning Alert and Response System (EWARS) and the Fiji Centre for Disease Control (FCDC) laboratory surveillance. Routine reporting of dengue cases is at the level of health facility and medical subdivision, not by residential location, and *Wolbachia* release zones do not align with medical subdivision boundaries. As such, a dengue case notified by a health facility within a release area may or may not actually reside in the release area, precluding the evaluation of dengue case occurrence with respect to the *Wolbachia* intervention. Environmental health teams do collect information on the residential location of dengue cases in the course of their case investigations and vector control response activities, but these operational databases were not available for the current analysis. The public health outcomes of the Fiji *w*Mel-*Ae. aegypti* deployments will thus be evaluated in a future analysis.

## Results

### *Wolbachia* deployment & introgression

**Fiji.** *w*Mel-infected adult *Ae. aegypti* were released in Suva, Nadi and Lautoka in 2018 and 2019. The number of weeks in which a release occurred in each area, an estimation of total mosquitoes released, mosquitoes released per km$^2$ and mosquitoes released per inhabitant are provided in S4 Table. Mosquito releases resulted in *w*Mel introgression into *Ae. aegypti* populations in 12 of 12 reporting areas in Suva, 5 of 6 reporting areas in Nadi and 5 of 5 reporting areas in Lautoka (Figs 1–3). In Suva, the prevalence of *w*Mel-infected *Ae. aegypti* oscillated in the Samabula and Suva City reporting areas after releases stopped in 2019 but nonetheless it was clearly established in the 2022 monitoring timepoint (Fig 1). In Nadi, the *w*Mel prevalence in trapped *Ae. aegypti* was greater than 80% in five of the six reporting areas at the time of last monitoring in January 2022 (Fig 2). No monitoring has been undertaken in the Denarau reporting area since November 2020, when there was strong evidence that *w*Mel was not established in this very small residential area. In Lautoka the *w*Mel prevalence in trapped *Ae. aegypti* was greater than 85% in all five of the reporting areas at the last monitoring visit in January 2022 (Fig 3). Collectively, these data demonstrate successful *w*Mel introgression in three of the most populous areas of the island of Viti Levu, Fiji.

**Vanuatu.** Port Vila is the capital city and largest and most populous urban area in Vanuatu. *w*Mel-infected adult *Ae. aegypti* were released in Port Vila over an ~5 month period in 2018/19. The number of weeks in which a release occurred in each area, an estimation of total mosquitoes released, mosquitoes released per km$^2$ and mosquitoes released per inhabitant are provided in S4 Table. At the time of last monitoring in May 2021 *w*Mel was established in ten

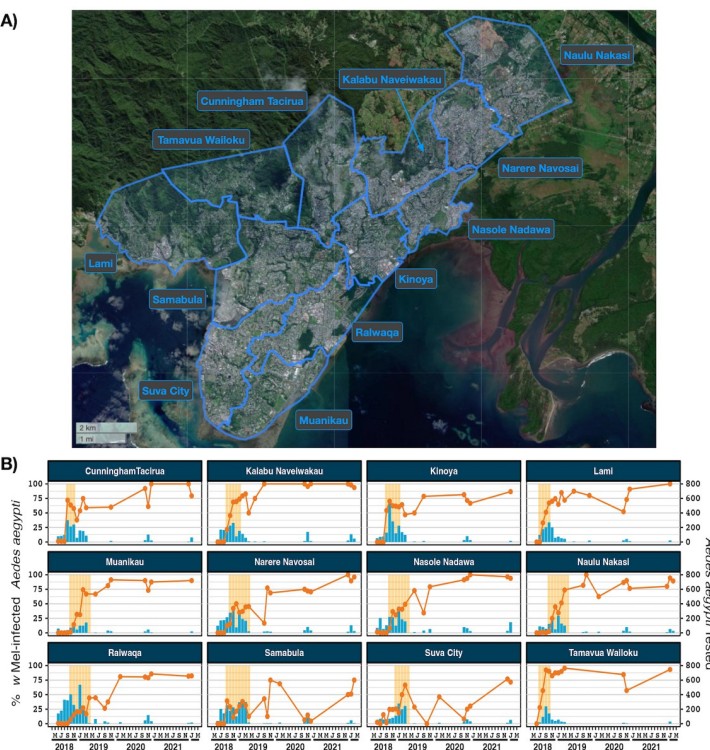

**Fig 1. *w*Mel introgression in 12 release areas in Suva, Fiji.** A) Suva and Lami, Fiji showing the 12 release zones. B) *w*Mel introgression. The line (left axis) represents the percent of *Ae. aegypti* tested that were infected with *w*Mel *Wolbachia*, between June 2018 and January 2022. The bars (right axis) indicate the number of *Ae. aegypti* tested. Results from reporting areas where there were less than five tested mosquitos have been omitted. Shaded orange areas indicate *w*Mel mosquito release times. Map produced in QGIS version 3.16.1 using boundaries aggregated from the enumeration area boundaries freely available from the Pacific Data Hub (https://pacificdata.org/data/dataset/2007_fji_phc_admin_boundaries) and OpenMapTiles basemap layer (https://openmaptiles.org/) with CARTO light design (https://carto.com/)).

of the twelve reporting areas, with five of these reporting areas having more than 95% of *Ae. aegypti* infected with *w*Mel (Fig 4). In the Mele Mele Maat reporting area *w*Mel was not established given the very low prevalence of *w*Mel detected in 2020 and not detected at all in May 2021. In the Bellvue Tassiriki reporting area it is unclear whether *w*Mel is established as *w*Mel prevalence results were highly variable due to low sample numbers of *Ae. aegypti* caught at each monitoring timepoint.

**Kiribati.** Tarawa atoll is the most populous island in Kiribati. wMel deployments were focused in South Tarawa, the area with the highest human population density. The scope of the South Tarawa project was reduced from eight to two reporting areas after approximately three months of mosquito releases. This decision was made when it became evident that *w*Mel introgression was unlikely to occur in all areas because insufficient *w*Mel-infected adult *Ae. aegypti* were being released relative to the abundance of wild-type mosquitoes. The two reporting areas that were prioritised for releases were Betio and Bairiki because 48% of the South Tarawa human population lived there. Mosquito releases in Betio and Bairiki occurred over an ~8 month period in 2018/19. The number of weeks in which a release occurred in each area, an estimation of total mosquitoes released, mosquitoes released per km$^2$ and mosquitoes released per inhabitant are provided in S3 Table.

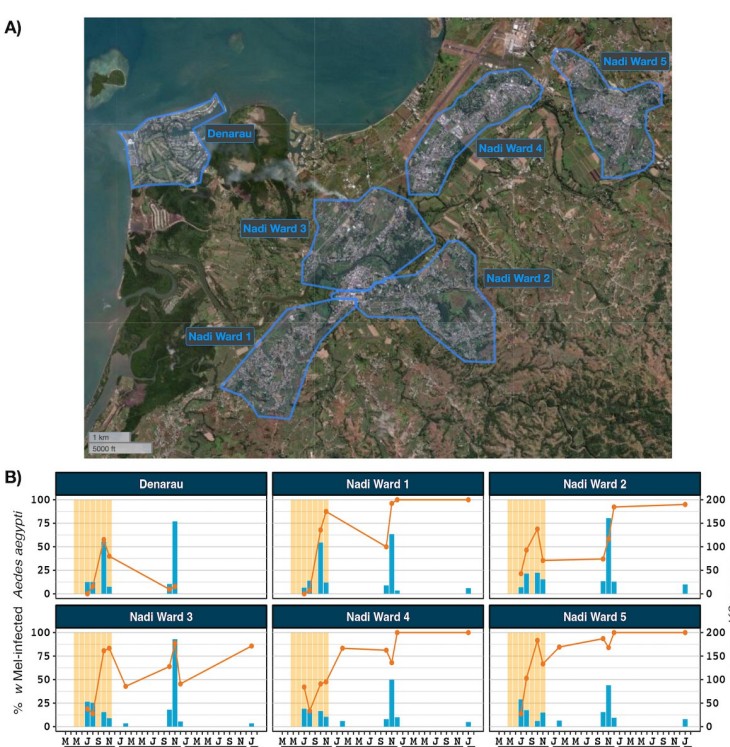

**Fig 2. wMel introgression in six release areas in Nadi, Fiji.** A) Nadi, Fiji showing the six release zones. B) *w*Mel introgression. The line (left axis) represents the percent of *Ae. aegypti* tested that were infected with *w*Mel *Wolbachia*, between July 2019 and January 2022. The bars (right axis) indicate the number of *Ae. aegypti* tested. Data points with less than five tested mosquitos have been omitted. Shaded orange areas indicate *w*Mel mosquito release times. Map produced in QGIS version 3.16.1 using boundaries aggregated from the enumeration area boundaries freely available from the Pacific Data Hub (https://pacificdata.org/data/dataset/2007_fji_phc_admin_boundaries) and OpenMapTiles basemap layer (https://openmaptiles.org/) with CARTO light design (https://carto.com/)).

Long term monitoring has demonstrated *w*Mel was established at a high prevalence in Betio, the most populous area of South Tarawa. In the Bairiki reporting area, *w*Mel prevalence was intermediate, being high on the western side most connected to Betio, but lower on the eastern side (Fig 5). BG traps (three traps total) from the eastern side had between 14.3–31.8% infected mosquitoes, whereas the western traps (seven traps total) had between 50–100% *Wolbachia*-infected mosquitoes.

**Impact of wMel Deployment on Aedes-transmitted Arboviral Disease Incidence.** Closure of international borders and internal restrictions on travel in response to the Covid19 pandemic in early 2020, until the gradual re-opening in 2022, will have impacted dengue epidemiology and case notification systems in all three countries. Against this backdrop, we report here the short-term epidemiological outcomes of *w*Mel establishment in Vanuatu and Kiribati. Data was not available for the evaluation of epidemiological outcomes in Fiji.

**Vanuatu.** Although sporadic dengue outbreaks have been reported in Vanuatu since the 1970s, with cases detected through passive hospital-based surveillance [32], consistent electronic records of suspected and laboratory-confirmed dengue cases have only been maintained by the Surveillance Unit of the Ministry of Health since 2016. Our evaluation of the public health impact of the Port Vila *w*Mel deployments is therefore limited by a short baseline period of only two years prior to releases of *w*Mel-infected *Ae. aegypti*. That baseline period includes a very large outbreak in 2017, with 1131 dengue cases reported in Port Vila, including 226

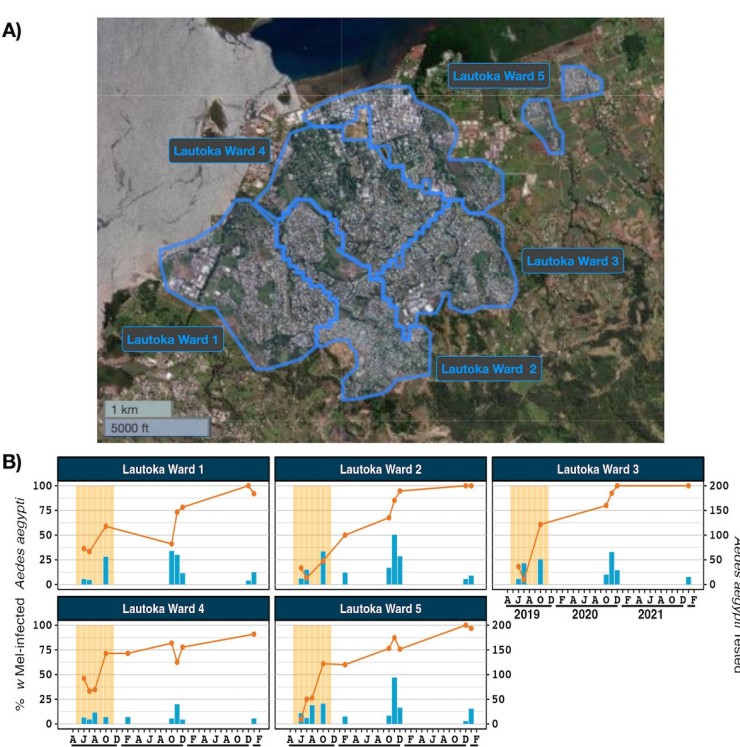

**Fig 3. *w*Mel introgression in five release areas in Lautoka, Fiji.** A) Lautoka, Fiji showing the five release zones. B) *w*Mel introgression. The line (left axis) represents the percent of *Ae. aegypti* tested that were infected with *w*Mel *Wolbachia*, between June 2019 and January 2022. The bars (right axis) indicate the number of *Ae. aegypti* tested. Data points with less than five tested mosquitos have been omitted. Shaded orange areas indicate *w*Mel mosquito release times. Map produced in QGIS version 3.16.1 using boundaries aggregated from the enumeration area boundaries freely available from the Pacific Data Hub (https://pacificdata.org/data/dataset/2007_fji_phc_admin_boundaries) and OpenMapTiles basemap layer (https://openmaptiles.org/) with CARTO light design (https://carto.com/)).

laboratory-confirmed cases (Fig 6). Comparatively few dengue cases have been reported in the years following the release of *w*Mel in Port Vila, though it is difficult to determine whether this represents a natural low in the interannual dengue cycle or a protective effect of *w*Mel against dengue virus transmission. Six laboratory-confirmed dengue cases (one NS1 positive and five IgM positive) were reported in 2020, with no hospitalisations (Fig 6). In 2021, 37 lab-confirmed dengue cases were reported but these cases were dispersed throughout the year (Fig 6).

**Kiribati.**   Prior to release of *w*Mel-infected *Ae. aegypti*, Kiribati experienced large dengue outbreaks in 2013/14 and 2018 with 255 and 528 suspected dengue cases (103 and 199 laboratory-confirmed [NS1 or IgM-positive] cases), respectively, and sustained transmission over several months (Fig 7). Kiribati, like some other Pacific Island countries, also experienced a large chikungunya outbreak in 2014/2015 [5], however electronic records of chikungunya laboratory diagnostics results are available only from 2017 (Fig 8). Suspected and laboratory-confirmed (IgM-positive) chikungunya cases were reported throughout 2017, after which time very few suspected cases and only a single laboratory-confirmed case have been reported. In the three years following the completion of *Wolbachia* releases in August 2019, a total of 764 suspected dengue cases (144 laboratory-confirmed) were notified in Kiribati (to August 2022), predominantly between February–July 2021 (Fig 7). An 'address' field included in the case notification data records a residential location for 78% of dengue cases notified since 2017. Fig 7

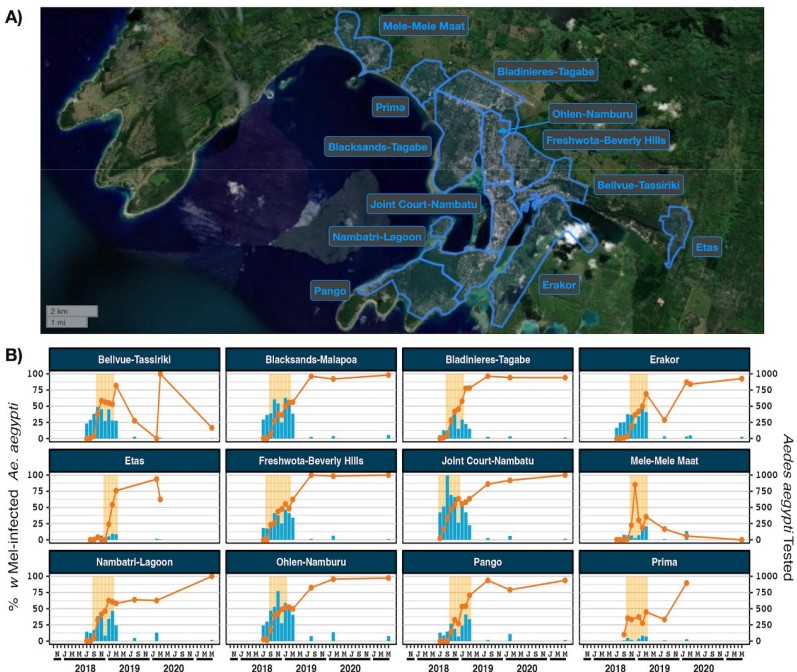

**Fig 4. *w*Mel introgression in 12 release areas in Port Vila, Vanuatu.** A) Port Vila, Vanuatu showing the 12 release areas. B) *w*Mel introgression. The line (left axis) represents the percent of *Ae. aegypti* screened that were infected with *w*Mel *Wolbachia*, between August 2018 and May 2021. The bars (right axis) indicate the number of *Ae. aegypti* tested. Data points with less than five screened mosquitos have been omitted. Shaded orange areas indicate *w*Mel mosquito release times. Map produced in QGIS version 3.16.1 using boundaries aggregated from the enumeration area boundaries freely available from the Pacific Data Hub (https://pacificdata.org/data/dataset/2016_vut_phc_admin_boundaries) and OpenMapTiles basemap layer (https://openmaptiles.org/) with CARTO light design (https://carto.com/)).

shows the monthly number of suspected and laboratory-confirmed dengue cases with a recorded address in the *Wolbachia*-treated areas of Betio and Bairiki, in non-release areas, and without a recorded address, before, during and after *Wolbachia* releases. In 2021–22 there were eight laboratory-confirmed dengue cases with a recorded address in Betio and seven in Bairiki, together accounting for 11% of all laboratory-confirmed cases in 2021–22. In the 2018 outbreak prior to *Wolbachia* releases, 19% (37/199) of laboratory-confirmed cases had a recorded address in Betio or Bairiki. No laboratory-confirmed chikungunya cases have been reported in Kiribati since the end of *Wolbachia* releases (Fig 8).

## Discussion

The *w*Mel strain of *Wolbachia* was durably established in local *Ae. aegypti* populations in the most populous areas of Fiji, Vanuatu and Kiribati as an adjunct to existing public health approaches to prevent and control dengue. More time will be needed to measure the public health outcomes of the *w*Mel deployments because of the episodic natural history of dengue outbreaks in these settings, and the current unavailability of dengue case notifications data at a spatial resolution aligned with the *w*Mel release areas, particularly in Fiji.

Widespread community acceptance is important for the effectiveness of any public health intervention and especially the release of *Wolbachia*-infected mosquitoes. WMP has developed a public acceptance model (PAM) [25] that is adaptable to local contexts and partner

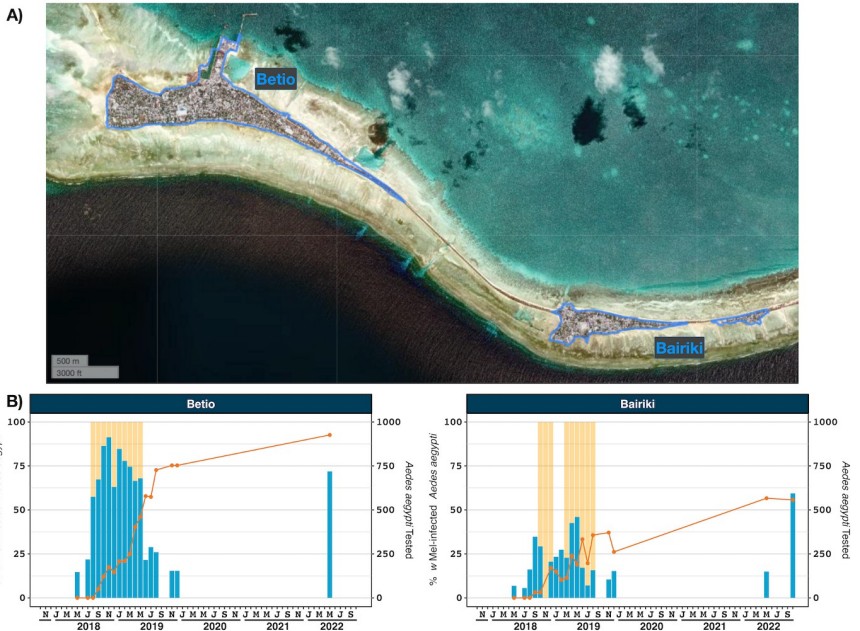

**Fig 5. *w*Mel introgression in two areas in South Tarawa, Kiribati. A) South Tarawa, Kiribati showing the two release areas: Betio (left) and Bairiki (right)**. B) Introgression of *w*Mel. The line (left axis) represents the percent of *Ae. aegypti* tested that were infected with *w*Mel Wolbachia, between May 2018 and December 2019. The bars (right axis) indicate the number of Ae. aegypti tested. Data points with less than five screened mosquitos have been omitted. Shaded orange areas indicate wMel mosquito release times. Map produced in QGIS version 3.16.1 using the enumeration area boundaries freely available from the Pacific Data Hub (https://pacificdata.org/data/dataset/2010_ kir_phc_admin_boundaries) and OpenMapTiles basemap layer (https://openmaptiles.org/) with CARTO light design (https://carto.com/).

requirements. Pre-release community survey results in all three participating countries demonstrated high level support for release of *w*Mel-infected mosquitoes, consistent with survey results from other international settings [19,22,25]. There were no material concerns from community members or stakeholders during or after the mosquito releases took place. A challenge for the future will be to refine how mass communications and community engagement is performed such that there is still strong support for deployment of *Wolbachia*-infected mosquitoes but with lower cost and with shorter time schedules.

We performed backcrossing to introgress nuclear genetic material from wild-type mosquitoes from each of Fiji, Vanuatu and Kiribati into the parental Cairns wMel line to create the three release lines. As a confirmatory test at the conclusion of backcrossing, each country line was challenged via intrathoracic injection with DENV-1-4 to demonstrate wMel-mediated resistance to DENV infection was retained. As a benchmark, we used the parental Cairns wMel line in the same experiments. We require that derivatives of the Cairns wMel line created by backcrossing to have similar phenotypes in response to intrathoracic challenge with DENV-1-4, i.e. significantly lower DENV burdens than their wild-type counterparts. As expected, this is what we observed with each of wMel Fiji, Vanuatu and Kiribati lines. The advantage of using intrathoracic challenge is that it can be standardised, is reproducible across time and requires fewer resources to perform than the alternative of oral challenges with viremic blood meals. *w*Mel introgression into an *Ae. aegypti* population can occur via deployment of adult mosquitoes (male and female) or eggs or both [19,22,23]. In Fiji, Vanuatu and Kiribati *w*Mel-infected mosquitoes were deployed as adults that had been reared in-country from

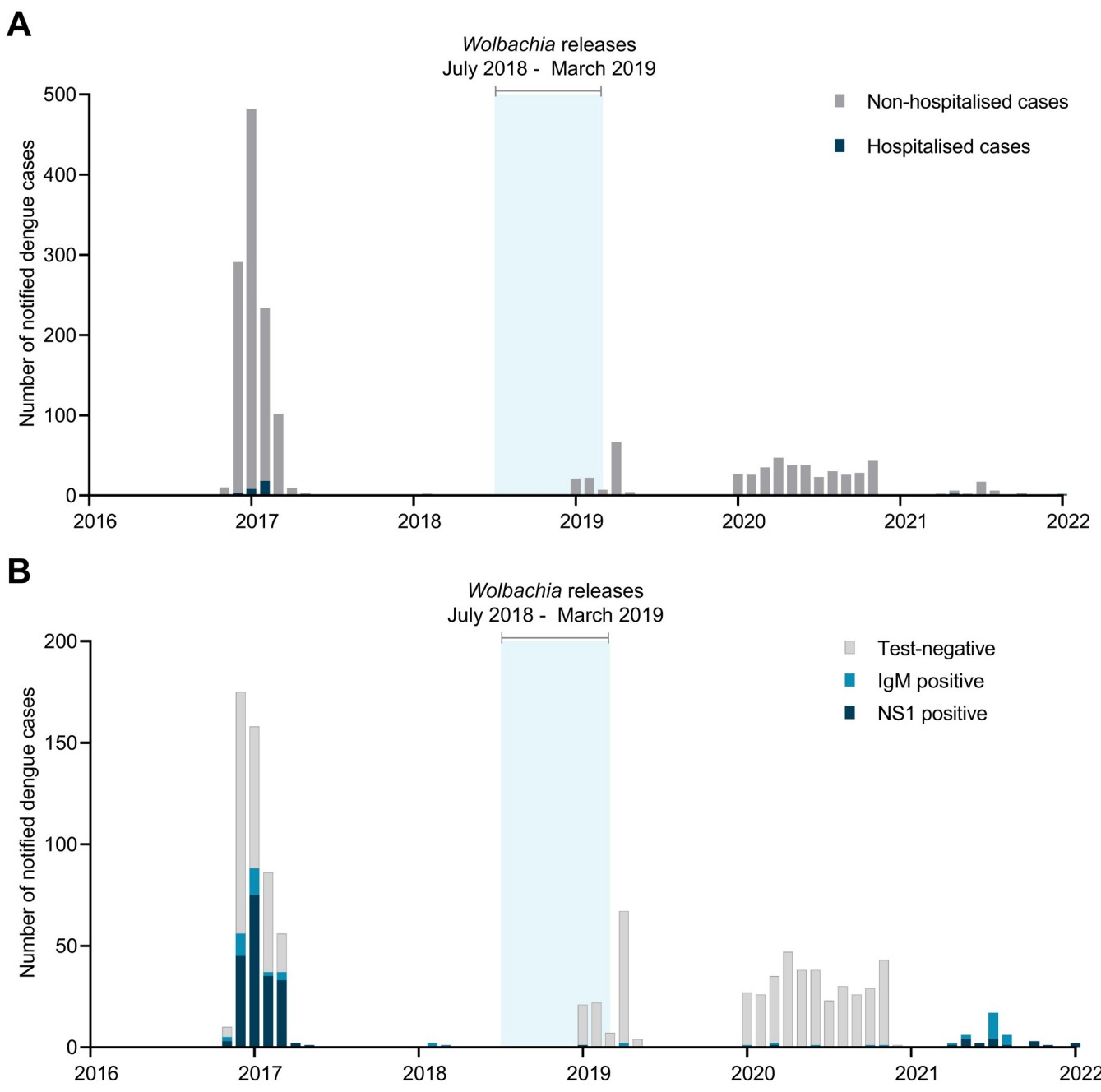

**Fig 6. Suspected dengue cases notified in Port Vila from January 2016 –January 2022 by (A) hospitalisation status and (B) diagnostic test result.** Blue shading indicates release period for *Wolbachia* (*w*Mel)-infected *Ae. aegypti*. Suspected dengue cases without any laboratory diagnostic testing are included in panel A, but excluded from panel B.

batches of eggs supplied every two weeks from an insectary facility in Melbourne, Australia [30]. Adult releases were supplemented in Kiribati with a small number of egg deployments. The decision to deploy adult mosquitoes, rather than eggs, was to avoid some of the human resource requirements needed for the careful deployment of eggs into the community. Notwithstanding the success of adult mosquito releases leading to *w*Mel introgression, the creation

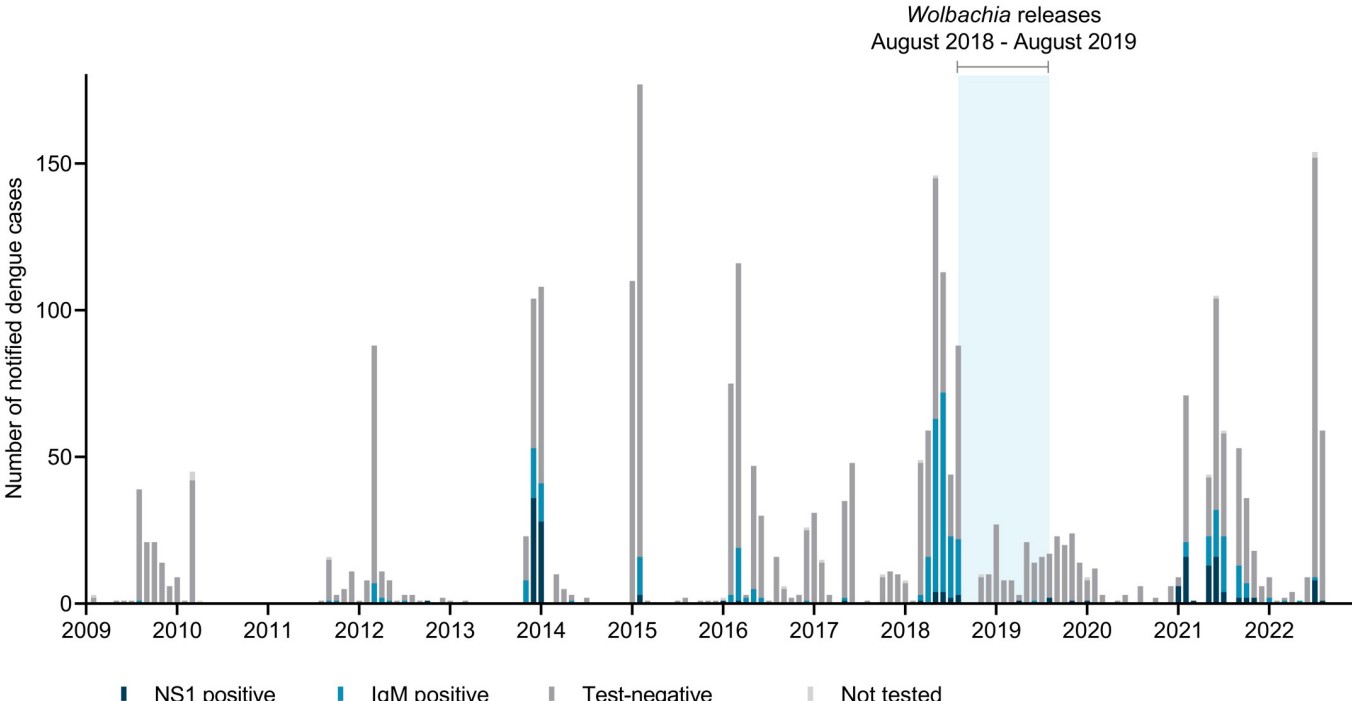

**Fig 7. Suspected dengue cases notified in Kiribati from January 2009 –August 2022 by diagnostic test result.** Blue shading indicates release period for *Wolbachia* (*w*Mel)-infected *Ae. aegypti*.

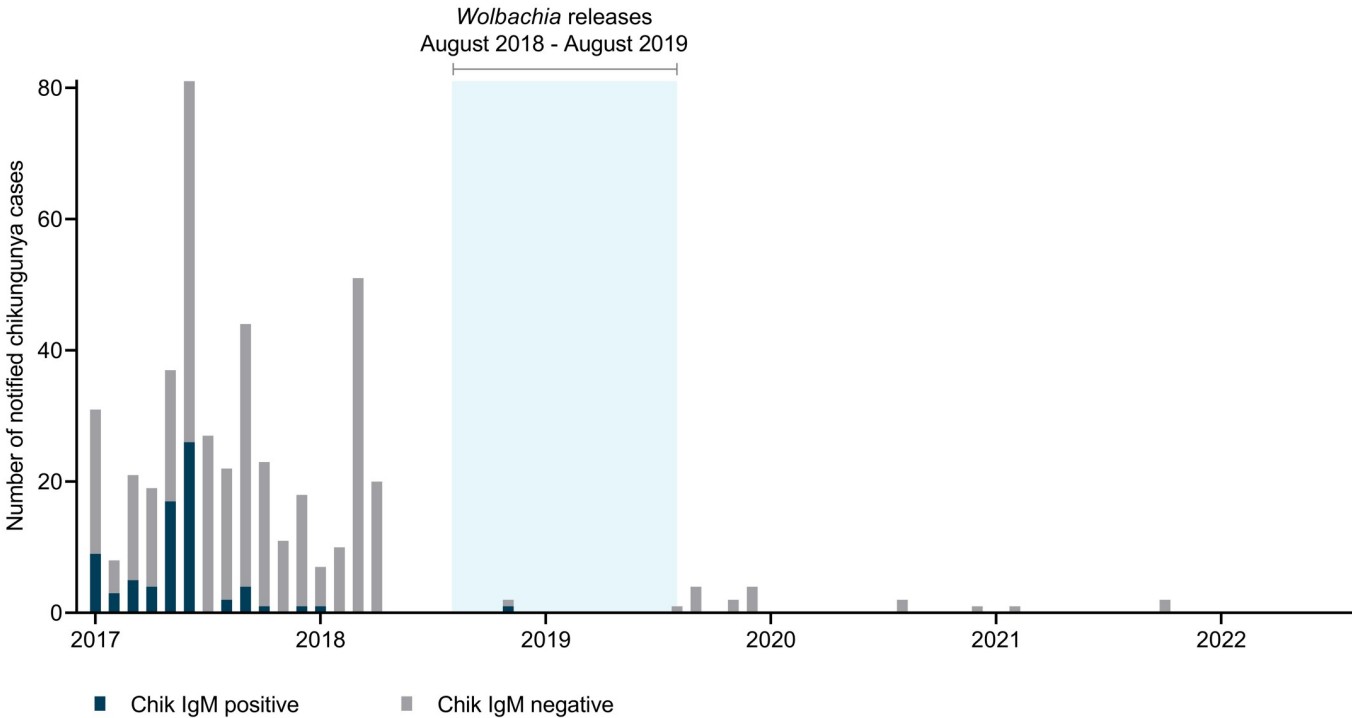

**Fig 8. Suspected chikungunya cases notified in Kiribati from January 2017 –August 2022.** Blue shading indicates release period for *Wolbachia* (*w*Mel)-infected *Ae. aegypti*.

of local insectaries in each country was challenged by logistical issues, staff training needs and problems maintaining control of insectary environmental conditions. In hindsight, the deployment of *w*Mel-infected mosquito eggs directly into the field after delivery from the Melbourne insectary might have been a more efficient operational model. Indeed, supply of *w*Mel-infected mosquito eggs to endemic countries from a small number of quality-assured regional insectary facilities is a simple model that should mean most endemic countries can access the intervention. Preserving hatch rates of transported *w*Mel-infected *Ae. aegypti* eggs and optimising simple and effective "last mile" delivery of eggs into communities should continue to be a focus of applied research efforts [33,34].

Long term monitoring demonstrates *w*Mel introgression has been achieved in nearly all of the reporting areas in Fiji and Vanuatu and in two of the planned eight areas in Kiribati. Collectively, this means that 429,000 people live in communities where *w*Mel establishment should afford protection from *Ae. aegypti*-transmitted arboviruses. Those areas where introgression clearly failed despite releases taking place (Mele-Mele Maat in Vanuatu and Denarau in the Central Division of Fiji) were small in human population size and geographically removed from the main release areas. The reasons for lack of introgression in these areas is unclear. In Denarau (Fiji), fortnightly fogging with insecticides by the private company that manages this residential estate and a lack of breeding sites in this well-maintained compound might have prevented wMel introgression. In Kiribati we chose to reduce the geographic scope of the project from eight areas to the two most populous areas (Betio and Bairiki). We did this for two reasons. First, it became evident that we were unable to reliably produce sufficient *w*Mel-infected eggs from the Melbourne insectary to meet the requirements for releases in all eight reporting areas. This was linked to generally lower fecundity of the Kiribati release line under mass production conditions versus Fiji and Vanuatu lines and our inability, because of insectary space constraints, to increase the number of Kiribati mosquitoes being reared for egg production. Second, the high baseline abundance of wild-type *Ae. aegypti* in South Tarawa (over seven times higher mean catch per trap per week than in Vanuatu and almost seventeen times higher than Fiji) necessitated a relatively higher release density of *w*Mel-infected mosquitoes than in Vanuatu or Fiji. The collective lesson from these projects was that pre-release entomological surveys to estimate the abundance of *Ae. aegypti* can usefully inform the density (mosquitoes/km$^2$) and duration for which *w*Mel-infected mosquitoes need to be released.

Vanuatu, Fiji and Kiribati each have their own entomological and epidemiological contexts that enable arbovirus transmission. In Vanuatu and Fiji the presence of the other competent mosquito species *Ae. albopictus* and *Ae. polynesiensis* in our release areas means that dengue virus transmission is still feasible despite *w*Mel establishment [35]. In Vanuatu and Kiribati, dengue was probably not endemic prior to *Wolbachia* deployments, i.e. not consistently present. Instead, historical dengue outbreaks were likely driven by viremic travellers who entered these countries and were bitten by a locally competent vector, sparking chains of transmission. These historical outbreaks only ended when a declining number of susceptible hosts, and perhaps vector control activities, reduced the effective reproduction number to below one. In Fiji, dengue was more likely to be endemic prior to *Wolbachia* deployments given the size of the country's population, existence of multiple populous urban communities and the volume of both domestic and inbound international travellers.

Although the infrastructure for infectious diseases surveillance differs between countries, there are several common challenges in Fiji, Vanuatu and Kiribati limiting the utility of routinely available dengue surveillance data for evaluating the health outcomes of the *w*Mel deployments. These include i) unknown specificity (i.e. the proportion of notified cases that are true cases), when suspected dengue cases are reported based on a syndromic case definition with absent or incomplete laboratory results; ii) unknown sensitivity (i.e. the proportion

of true cases that go undetected), due to variable access to health facilities and reporting only by sentinel facilities and large public hospitals; iii) incomplete or absent information on cases' residential location, precluding the reliable classification of cases as residing within or outside of a *Wolbachia* release area; iv) in Fiji and Vanuatu, changes to disease surveillance processes over time resulting in only a short pre-intervention baseline period with data comparable to post-intervention; v) in Fiji, parallel syndromic, laboratory-based and hospital-based surveillance systems that each provide a partial but unintegrated view of dengue disease burden, and which have complex data custody arrangements that have currently precluded the use of these data for evaluating the outcomes of the Fiji *w*Mel deployments. The incompleteness of residential location information in case notifications is a major barrier to the utility of these data not only for the evaluation of interventions as described here, but also for informing timely and effective public health responses.

Notwithstanding these limitations, routine disease surveillance data provides a pragmatic and readily available data source for evaluating the real-world effectiveness of public health interventions like *Wolbachia* that are implemented programmatically, and has the benefit of avoiding the implementation of parallel systems that risk burdening an already constrained system. In Vanuatu, the lack of a comparative untreated population; a short baseline period with comparable data that included only one large dengue outbreak pre-intervention; and the unavailability of dengue surveillance data for 2022 due to an interruption to government information systems and other public health priorities, meant that no conclusions can yet be made on the public health outcomes of the *w*Mel deployments in Port Vila. In Kiribati, an uptick in reported dengue cases in 2021 and 2022 compared to the previous two years included a small number of cases with a recorded address within the *w*Mel-treated areas of South Tarawa. *w*Mel was established at an intermediate-to-high prevalence in the two South Tarawa release areas by 2022. It is not possible to determine whether the notified dengue cases arose from local DENV transmission by remaining wild-type *Ae. aegypti* populations or acquired their infections elsewhere outside the relatively small (~2 km$^2$) *w*Mel release areas. In Fiji, a future evaluation of the long-term public health outcomes of the *w*Mel deployments will require integration across routine disease surveillance and operational vector control datasets, in order to determine the spatial distribution of dengue cases with respect to *w*Mel-treated areas.

The closure of international borders and domestic travel restrictions following the emergence of SARS-CoV-2 in early 2020, within a year of *w*Mel deployments finishing, will have impacted the local epidemiology of dengue in all three countries. The effective absence of inbound travellers is likely to have diminished the risk of dengue outbreaks in non-endemic Vanuatu and Kiribati, and disruptions to healthcare systems and changes in care-seeking, testing and reporting behaviours have affected notifiable disease reporting across geographies [36,37]. With the lifting of pandemic restrictions and resumption of domestic and international travel throughout the Pacific in 2022, continued monitoring of dengue and chikungunya case incidence is needed to strengthen the evidence for the public health impact of the *w*Mel deployments in Pacific communities.

The *Wolbachia* method is an egalitarian public health intervention. Once established, all residents within the target area are equally protected regardless of age, gender or socioeconomic background. Although the deployment of the *Wolbachia* method frequently benefits from community participation, it is not a necessity. However, a successful release does require community authorisation. Once established in a target *Ae. aegypti* population, *Wolbachia* is self-sustaining and requires no ongoing behaviour change or public participation [38,39], unlike other vector control activities like environmental management that require sustained buy-in from communities [40]. The *w*Mel deployments in the three PICs were targeted to the most populous urban centres, where the relatively high population density and baseline

dengue incidence make the intervention most cost-effective [41]. Dengue transmission occurs also in other smaller population centres in Fiji [42], Vanuatu and Kiribati, however future expanded *w*Mel deployments would need to consider cost-benefit. The degree to which *w*Mel-mediated suppression of DENV transmission in urban centres may indirectly reduce transmission in peripheral areas through reduced viral flow and seeding of outbreaks is also an unanswered question that warrants future research.

*Wolbachia*-infected *Ae. aegypti* are an additional in the public health toolbox for dengue control. As this work shows, deployments across numerous geographical locations can be achieved from centralised mosquito production facilities and local teams formed by partnerships with Ministries of Health. Further long-term monitoring will enable a fuller picture of the public health outcomes of *Wolbachia* deployments in these three Pacific Island countries.

## Supporting information

**S1 Fig. *Wolbachia*-mediated reduction in DENV genome copy number per mosquito.** All mosquitoes were aged for 6–7 days prior to intrathoracic injection with DENV. Fifty mosquitoes were used for each data point but some died prior to testing (S3 Table). DENV copy number was determined 7 days post injection using qRT-PCR. All *w*Mel *Ae. aegypti* lines had a significant reduction in DENV viral RNA concentration (Wilcoxon rank-sum est). A) Fiji release strain, Fij-*w*Mel, and wild-derived control, Fij-WT,vector competence. B) Vanuatu release strain, Van-*w*Mel, and wild-derived control, Van-WT, vector competence. C) Kiribati release strain, Kir-*w*Mel, and wild-derived control, Kir-WT, vector competence. D) Australian Cairns strain, Aus-*w*Mel, and tetracycline cured control, Aus-TET, vector competence. Data are shown as the median DENV copies per mosquito (thick line) ± interquartile ranges (box), extended by the whiskers indicating 1.5× the interquartile range, with dots indicating outliers. Individual data points are included as smaller partially opaque points. Data from uninfected mosquitoes are not included in the median estimates (S3 Table).
(TIFF)

**S2 Fig. Insecticide Resistance (IR) Profiles of Release Strains determined by WHO Biosaay.** A) Fiji release strain IR profile. B) Vanuatu release strain IR profile. C) Kiribati release strain IR profile. Each data point is the mean of five biological replicates (± s.d.) using approximately 20 mosquitoes per replicate.
(TIFF)

**S3 Fig. Release & monitoring of *w*Mel-infected *Ae. aegypti* within 12 areas of Suva and Lami, Fiji.** Each release area was divided into a grid with 100 x 100 meter squares. Grid squares lacking mosquito releases were omitted. Release gradient was determined by using GPS coordinates of each release event and assigning the number of *w*Mel-infected mosquitos to a corresponding grid square. Monitoring numbers were determined in the same way. Map produced in QGIS version 3.16.1 using boundaries aggregated from the enumeration area boundaries freely available from the Pacific Data Hub (https://pacificdata.org/data/dataset/2007_fji_phc_admin_boundaries) and OpenMapTiles basemap layer (https://openmaptiles.org/) with CARTO light design (https://carto.com/)).
(PNG)

**S4 Fig. Release & monitoring of *w*Mel-infected *Ae. aegypti* within six areas of Nadi and five areas of Lautoka, Fiji.** Each release area was divided into a grid with 100 x 100 meter squares. Grid squares lacking mosquito releases were omitted. Release gradient was determined by using GPS coordinates of each release event and assigning the number of *w*Mel-infected mosquitos to a corresponding grid square. Monitoring numbers were determined in the same way.

Map produced in QGIS version 3.16.1 using boundaries aggregated from the enumeration area boundaries freely available from the Pacific Data Hub (https://pacificdata.org/data/dataset/2007_fji_phc_admin_boundaries) and OpenMapTiles basemap layer (https://openmaptiles.org/) with CARTO light design (https://carto.com/)).
(PNG)

**S5 Fig. Release & monitoring of *w*Mel-infected *Ae. aegypti* within 12 areas of Port Vila, Vanuatu.** Each release area was divided into a grid with 100 x 100 meter squares. Grid squares lacking mosquito releases were omitted. Release gradient was determined by using GPS coordinates of each release event and assigning the number of *w*Mel-infected mosquitos to a corresponding grid square. Monitoring numbers were determined in the same way. Map produced in QGIS version 3.16.1 using boundaries aggregated from the enumeration area boundaries freely available from the Pacific Data Hub (https://pacificdata.org/data/dataset/2016_vut_phc_admin_boundaries) and OpenMapTiles basemap layer (https://openmaptiles.org/) with CARTO light design (https://carto.com/)).
(PNG)

**S6 Fig. Release & monitoring of *w*Mel-infected *Ae. aegypti* within two areas of South Tarawa, Kiribati.** Each release area was divided into a grid with 100 x 100 meter squares. Grid squares lacking mosquito releases were omitted. Release gradient was determined by using GPS coordinates of each release event and assigning the number of *w*Mel-infected mosquitos to a corresponding grid square. Monitoring numbers were determined in the same way. Map produced in QGIS version 3.16.1 using the enumeration area boundaries freely available from the Pacific Data Hub (https://pacificdata.org/data/dataset/2010_kir_phc_admin_boundaries) and OpenMapTiles basemap layer (https://openmaptiles.org/) with CARTO light design (https://carto.com/)).
(PNG)

**S7 Fig. Total (A) and laboratory-confirmed (B) dengue cases notified in Kiribati from January 2017 –August 2022 by recorded location of residence.** Cases with an 'address' location recorded were classified either as resident in the *Wolbachia*-release areas of Betio and Bairiki or in non-release areas (all locations other than Betio and Bairiki). Laboratory-confirmed dengue cases include those with a positive dengue NS1 and/or IgM diagnostic test result recorded. Blue shading indicates the *Wolbachia* release period.
(TIF)

**S1 Table. Regulatory Permits for Pacific Releases.**
(DOCX)

**S2 Table. Pre-release Mosquito Strain Health Checks.**
(DOCX)

**S3 Table. DENV Prevalence in *Wolbachia*-infected Mosquitoes.**
(DOCX)

**S4 Table. *Wolbachia* (*w*Mel)-infected *Ae. aegypti* Mosquito Release Numbers.**
(DOCX)

## Acknowledgments

We would like to thank all members of the World Mosquito Program, our partners and the communities in Fiji, Vanuatu and Kiribati for their tireless efforts in making this novel public health intervention a success.

## Author Contributions

**Conceptualization:** Cameron P. Simmons, Peter A. Ryan, Katherine L. Anders, Scott L. O'Neill.

**Data curation:** Tibor Frossard, Jai A. Denton.

**Formal analysis:** Cameron P. Simmons, Anaseini Ratu, Jai A. Denton, Stephanie K. Tanamas, Katherine L. Anders.

**Funding acquisition:** Cameron P. Simmons, Peter A. Ryan, Scott L. O'Neill.

**Investigation:** Cameron P. Simmons, Lekon Tagavi, Len Tarivonda, Timothy Quai, Raynelyn Tavoa, Tebikau Noran, Erirau Manikaoti, Lavinia Kareaua, Tabomoa Tinte Abwai, Dip Chand, Vineshwaran Rama, Vimal Deo, Kharishma Karti Deo, Aminiasi Tavuii, Wame Valentine, Raviikash Prasad, Eremasi Seru, Leikitah Naituku, Anaseini Ratu, Mark Hesketh, Nichola Kenny, Sarah C. Beebe, Anjali A. Goundar, Andrew McCaw, Molly Buntine, Geoff Wilson, Johanna M. Duyvestyn, Etiene C. Pacidônio, Heather A. Flores, Jai A. Denton, Peter Kyrylos, Yi Dong, Katherine L. Anders, Scott L. O'Neill.

**Methodology:** Cameron P. Simmons, Peter A. Ryan, Katherine L. Anders, Scott L. O'Neill.

**Project administration:** Mark Hesketh, Nichola Kenny, Sarah C. Beebe, Anjali A. Goundar, Andrew McCaw, Molly Buntine, Ben Green, Jeremie R. L. Gilles, D. Albert Joubert, Geoff Wilson, Le Quyen Duong, Jean B Bouvier, Darren Stanford, Carolyn Forder, Heather A. Flores, Natalie Wittmeier, Kate Retzki, Peter A. Ryan, Ruth Smithyman, Stephanie K. Tanamas, Peter Kyrylos, Yi Dong, Anam Khalid, Lauren Hodgson, Katherine L. Anders, Scott L. O'Neill.

**Supervision:** Cameron P. Simmons, Timothy Quai, Tebikau Noran, Vimal Deo, Kharishma Karti Deo, Aminiasi Tavuii, Anjali A. Goundar, Jeremie R. L. Gilles, Geoff Wilson, Ruth Smithyman.

**Visualization:** Tibor Frossard, Jai A. Denton, Stephanie K. Tanamas, Katherine L. Anders.

**Writing – original draft:** Cameron P. Simmons, Jai A. Denton, Stephanie K. Tanamas, Katherine L. Anders.

**Writing – review & editing:** Cameron P. Simmons, Wesley Donald, Dip Chand, Anaseini Ratu, Heather A. Flores, Jai A. Denton, Katherine L. Anders, Scott L. O'Neill.

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
