## [Decision Letter · Decision Letter 0]

17 Aug 2023

Dear Dr. Simmons,

Thank you very much for submitting your manuscript "Successful introgression of *w*Mel *Wolbachia* into *Aedes aegypti* populations in Fiji, Vanuatu and Kiribati" for consideration at PLOS Neglected Tropical Diseases. As with all papers reviewed by the journal, your manuscript was reviewed by members of the editorial board and by several independent reviewers. In light of the reviews (below this email), we would like to invite the resubmission of a significantly-revised version that takes into account the reviewers' comments. 

We cannot make any decision about publication until we have seen the revised manuscript and your response to the reviewers' comments. Your revised manuscript is also likely to be sent to reviewers for further evaluation.

Sincerely,

Mariangela Bonizzoni

Academic Editor

Audrey Lenhart

Section Editor

Reviewer's Responses to Questions

**Key Review Criteria Required for Acceptance?**

**Methods**

-Are the objectives of the study clearly articulated with a clear testable hypothesis stated?

-Is the study design appropriate to address the stated objectives?

-Is the population clearly described and appropriate for the hypothesis being tested?

-Is the sample size sufficient to ensure adequate power to address the hypothesis being tested?

-Were correct statistical analysis used to support conclusions?

-Are there concerns about ethical or regulatory requirements being met?

Reviewer #1: In relation to field monitoring survey, I reckon it is important to provide estimates of the Ae. aegypti population size prior and during the releases in order to assess the population density for efficient release design (as stated by the authors in the Discussion), but also for assessing the impact of the releases at level of local mosquito population size. 

This also represents an important aspect which should be assessed in the post-release survey of local communities (for example, the perceived estimate of mosquitoes circulating in release areas). In addition, data on the composition of the Aedes population (i.e. other sympatric species, like Ae. albopictus, Ae. polynesiensis) would also be very informative and useful to interpret the epidemiological impact of the releases. This monitoring can be done by placing ovitraps on release sites and by performing a comparative analysis with data from control sites. Have the authors collected this data?

Could you comment on the choice of not including matching control (i.e. where no release of wMel-carrying Ae. aegypti occur) sites, potentially surrounding release sites. Is there a rational for this? Comparative epidemiological data and mosquito population density data from these sites would have been very important in this study.

Reviewer #2: Although the authors give a clear outline of the release sites in the three Islands of choice it is unclear to me why they did not included a detailed outline of control sites or was it the case that none were included? The benefit of geographically distinct control sites, where no mosquitoes are released and area is similar to release sites i.e. population structure of vector and host, geographical location, terrain, urban/rural etc, is that a more direct evaluation of the impact of wMel releases can be obtained. For example, direct comparison of dengue cases in release vs control sites could be made (although I appreciate this may not always be possible as stated by the authors there is often incomplete residential data for reported infections). 

Can the authors comment on why when carrying out fitness cost there was no data obtained from corresponding parental lines? This at the very least would determine if the introgression of wMel into the local background or the genetic background of the local mosquitoes in general have fitness issues. For example, the apparent low fecundity rate of Van-wMel (38.92 ± 39.79 (s.d.)), if this true for the parental Van line?

Reviewer #3: 1. What is minimum recommendation for mosquito sample size post-release monitoring?

2. What is minumum trapping design to allow robust monitoring?

3. Was post monitoring done in any particular season - would that matter?

4. Is thoracic challenge sufficient for these kinds of studies?

5. Any idea of approximate release ratio Wolbach : wildtype? Why no pre-intervention baseline? This would be standard practice for any biological release and control program.

**Results**

-Does the analysis presented match the analysis plan?

-Are the results clearly and completely presented?

-Are the figures (Tables, Images) of sufficient quality for clarity?

Reviewer #1: Datasets from the post-release monitoring are limited in some sites (such as Suva, Vanuatu), compared to others – is this related to a very low population density in those areas or to the applied collection methods? Why the authors have opted for adult collections (with traps or aspirators) over using ovitraps (this would have probably increased the number of individuals to be tested)?

I suggest to also include the phenotypic data of the original wMel line (Cairns Ae. aegypti wMel-infected line) and the wild-type line sharing the same genetic background, used for introgressions, where appropriate depending on the assay. This is important for comparing possible costs related to the genetic background. In relation to this point, reduced fecundity and hatching rate were observed in the Van-wMel line, compared to the other genetic backgrounds (Fecundity: 38.92 ± 39.79 (s.d.) Hatching: 49 females;33.76 ± 39.63 (s.d.); 21 females with 0% hatch rate). Does the Van-WT line have a similarly reduced fecundity and fertility? In the Discussion, the authors associate the failure of introgression in some areas and the reduction of the geographic scope of the project with the low fecundity of the Kiribati line (“This was linked to generally lower fecundity of the Kiribati release line versus Fiji and Vanuatu lines”). However, looking at S2 Table. Pre-release Mosquito Strain Health Checks, this line showed a relatively higher fecundity (80.85 ± 29.65 (s.d.) and hatch rate(57.89 ± 33.21% (s.d.)). Can the authors discuss this point?

Reviewer #2: Can the authors comment on the low number of mosquitoes used to assess the long-term level of wMel within the populations? 

The authors do not present data on Wolbachia density within each line and in comparison to the parental wMel line. Can the authors comment on if there are differences in Wolbachia density between the different genetic backgrounds? 

 It would be beneficial for the authors to discuss why DENV challenges were carried out by intrathoracic injection and not via blood meal. Given the reduction in follow up data due to the Sars-CoV-2 pandemic with regards to dengue cases, clearly showing that each introgressed background line reduces dissemination of DENV would have made a stronger paper. To this end the authors should discuss why field mosquitoes were not collected from release sites and further challenged with DENV to show that the lines retained the ability to block the virus, this would have greatly improved the dataset particularly in light of the interruption to data analysis caused by the pandemic. 

In figure S1 The authors state that 50 mosquitoes were used for each datapoint. However the data points are unclear in many of the box plots. Can the authors comment on this? How many data points were used per group?

Reviewer #3: 1. Fig 1-5. Address why it is not possible to give any estimate of variance around % wolbachia infected.

2. Why not include Fiji dengue data? Limitations re: spatial and temporal accuracy can't be that different to the other countries?

**Conclusions**

-Are the conclusions supported by the data presented?

-Are the limitations of analysis clearly described?

-Do the authors discuss how these data can be helpful to advance our understanding of the topic under study?

-Is public health relevance addressed?

Reviewer #1: The aim of the paper was to describe the introgression of Wolbachia wMel into the local populations of Ae. aegypti. The characterization of the basic life hostly parameters and viral blockage capacity of the newly introgressed lines were assessed. Although with some limitations, data on the field monitoring of Wolbachia prevalence in recapture mosquitoes were reported. Therefore, the overall conclusions of the study are supported by the data presented. Epidemiological data are not solid enough for drawing robust conclusions of the impact of the releases, however the public health outcome was not intended as the main focus of the study.

Nevertheless, I suggest to address the following points in the discussion section:

Phenotypic stability and viral blockage of some variants of wMel is also dependent on environmental conditions, such as temperature. Therefore, it is important to include details about the climate of the release areas and to evaluate whether such susceptibility could be a potential factor contributing to the reduced establishment and invasion ability of wMel-carrying mosquitoes in some areas.

Although being aware that viral infections via intra- thoracic injections and viral quantification using qPCRs on whole bodies are operationally easier for performing viral challenges using four different dengue serotypes, they represent a proxy with substantial limitations and biases for assessing the true ability of the mosquito lines to transmit arboviruses. Therefore, I recommend that these limitations of the study should be at least aknowledged and addressed in the text.

Reviewer #2: When discussing potential reasons for failure of introgression the authors fail to discuss that wMel density and maternal transmission can be affected by environmental factors such as high temperatures. Although not necessarily the case here, it would be prudent to at least discuss it. 

The authors state that they reduced the number of release sites in Kiribati due to several reasons one being the reduced fecundity of the introgressed background line used in this area. However, the fecundity data presented does not appear to align with this: 80.85 ± 29.65 (s.d.)) for the Kir-wMel compared to say 38.92 ± 39.79 (s.d.) for the Van-wMel line. This would suggest that in fact the fecundity for Kir-wMel was in fact much better than for Van-wMel. Further to this the standard deviation for the Van-wMel line is extremely high. Can the authors comment on this and discuss if there are highly levels of variation and an abnormal distribution of data? If so, it may be more advisable to use the median rather than the mean and to state the range rather than the standard deviation.

The authors have taken the time to evaluate the insecticide resistance phenotypes which is important and relevant to the future of release programs such as this. However, they fail to discuss this in the manuscript. It would be beneficial to add a few sentences on this, particularly with respect to issues described in the Denarau area of Fiji where the authors state “active vector control activities in a largely expatriate community might have impeded wMel introgression.” Is this insecticide fogging? Larval management? The authors should also comment on the rationale behind the use of the Rockerfeller line and the exclusion of the parental Carins line from these experiments.

Reviewer #3: 1. What are the economics of dengue in these hypoendemic countries?

2. What are the costs of Wolbachia release?

3. What is the future of the Pacific program post provision of funds from a high income government?

4. Could LAMP not be transferred to local health authorities?

5. Did COVID affect post-release monitoring? Can we really compare 2019 and before with 2020, 2021? 

6. Was lower fecundity in Kiribas strain real, or an artifact of lab rearing / selection?

**Editorial and Data Presentation Modifications?**

Reviewer #1: (No Response)

Reviewer #2: (No Response)

Reviewer #3: 1. Is Fig 8 necessary - what does it tell us? 

2. Standardise backcrossing descriptions in S2 Table.

3. Standardise legends, time scales etc in Fig 6 and 7

**Summary and General Comments**

Reviewer #1: The manuscript by Simmons et colleagues describes the introgression and establishment of populations of wMel-carrying Aedes aegypti in five countries of the Pacific island region for dengue control interventions. The program of public engagement activities and local communities involvement carried out prior and during the releases appear robust and properly assessed. The study of the basic phenotypic parameters of the newly introgressed lines is extensive, including viral challenges for assessing mosquito vector competence using four different dengue serotypes. 

Outcomes from Wolbachia-based field interventions are important not only for expanding the application of Wolbachia in reducing dengue incidence, but also for providing significative case-based insights for releases optimization.

Considering the limitations and challenges imposed by the COVID-19 pandemics, the authors managed to provide a good report showing long-term monitoring data on the establishment of the released population, including a framework of epidemiological data of dengue and Chikungunya based on passive case monitoring (although most of them are not confirmed cases).

Reviewer #2: The manuscript by Simmons et al presents a timely and important evaluation of mosquito control programs, releasing Wolbachia strain wMel into wild populations to limit the spread of arboviruses such as dengue. The manuscript described in detail the necessary steps required to develop and implement a successful integration protocol in different geographical areas and is very well written. The authors do an excellent job of discussing the difficulties and breadth of community engagement required for such release programs to be successful. The authors present showing the successful introgression of wMel carrying mosquitoes into native wild population and that this appears to be sustained. They also show very preliminary data on the effect this introgression is having on dengue incidents, although as the authors point out this was severely hindered by the Sars-CoV-2 pandemic. 

The paper is very well written and really shows the in-depth procedures required for successful release programs. The authors also do a great job of discussing the potential barriers and issues that may arise as the number of release sites increase. 

Although a timely and important study showing the use of Wolbachia control strategies in the Pacific Island communities, there are several concerns I feel the authors must address before the paper can be accepted for publication.

Reviewer #3: A well-written, generally descriptive paper re: developments with Wolbachia release programs in the Pacific. It is of interest as an update on activities and because WHO is soon to endorse / PQ list the intervention. Title does not reflect fact that data is largely descriptive rather than statistically robust, and that releases failed in small number of areas.

PLOS authors have the option to publish the peer review history of their article (what does this mean?). If published, this will include your full peer review and any attached files.

Reviewer #1: No

Reviewer #2: No

Reviewer #3: No
---

## [Decision Letter · Decision Letter 1]

25 Feb 2024

Dear Dr. Simmons,

We are pleased to inform you that your manuscript 'Successful introgression of *w*Mel *Wolbachia* into *Aedes aegypti* populations in Fiji, Vanuatu and Kiribati' has been provisionally accepted for publication in PLOS Neglected Tropical Diseases.

Best regards,

Mariangela Bonizzoni

Academic Editor

Audrey Lenhart

Section Editor

Reviewer's Responses to Questions

**Key Review Criteria Required for Acceptance?**

**Methods**

-Are the objectives of the study clearly articulated with a clear testable hypothesis stated?

-Is the study design appropriate to address the stated objectives?

-Is the population clearly described and appropriate for the hypothesis being tested?

-Is the sample size sufficient to ensure adequate power to address the hypothesis being tested?

-Were correct statistical analysis used to support conclusions?

-Are there concerns about ethical or regulatory requirements being met?

Reviewer #1: (No Response)

Reviewer #2: The Authors have addressed all my concerns

**Results**

-Does the analysis presented match the analysis plan?

-Are the results clearly and completely presented?

-Are the figures (Tables, Images) of sufficient quality for clarity?

Reviewer #1: (No Response)

Reviewer #2: The Authors have addressed all my concerns

**Conclusions**

-Are the conclusions supported by the data presented?

-Are the limitations of analysis clearly described?

-Do the authors discuss how these data can be helpful to advance our understanding of the topic under study?

-Is public health relevance addressed?

Reviewer #1: (No Response)

Reviewer #2: The Authors have addressed all my concerns

**Editorial and Data Presentation Modifications?**

Reviewer #1: (No Response)

Reviewer #2: The figures in the PDF are a little on the blurry side and this should be addressed in the editorial editing before publication.

**Summary and General Comments**

Reviewer #1: The authors addressed most of my comments and edited the manuscript accordingly.

Reviewer #2: I would like to thank the authors for taking the time to clearly and concisely address all concerns I raised in my initial review. All concerns have been addressed and I would like to congratulate the authors for a timely and accurate representation of Wolbachia deployment strategies.

PLOS authors have the option to publish the peer review history of their article (what does this mean?). If published, this will include your full peer review and any attached files.

Reviewer #1: No

Reviewer #2: No

---

## [Editor Report · Acceptance letter]

10 Mar 2024

Dear Dr. Simmons,

We are delighted to inform you that your manuscript, "Successful introgression of *w*Mel *Wolbachia* into *Aedes aegypti* populations in Fiji, Vanuatu and Kiribati," has been formally accepted for publication in PLOS Neglected Tropical Diseases.

Best regards,

Shaden Kamhawi

co-Editor-in-Chief

Paul Brindley

co-Editor-in-Chief
